# The Anxiolytic Activity of *Schinus terebinthifolia* Leaf Lectin (SteLL) Is Dependent on Monoaminergic Signaling although Independent of the Carbohydrate-Binding Domain of the Lectin

**DOI:** 10.3390/ph15111364

**Published:** 2022-11-07

**Authors:** Bárbara Raíssa Ferreira de Lima, Leydianne Leite de Siqueira Patriota, Amanda de Oliveira Marinho, Jainaldo Alves da Costa, Thiago Henrique Napoleão, Michelle Melgarejo da Rosa, Patrícia Maria Guedes Paiva

**Affiliations:** 1Department of Biochemistry, Biosciences Center, Federal University of Pernambuco, Recife 50670-901, PE, Brazil; 2Therapeutic Innovation Research Center Suely Galdino (NUPIT), Federal University of Pernambuco, Recife 50670-420, PE, Brazil

**Keywords:** anxiety, elevated plus maze, lectin, mice models, open field

## Abstract

The potential of plant lectins (carbohydrate-binding proteins) for the treatment of neurological disorders such as anxiety and depression has started to be reported in the last few years. *Schinus terebinthifolia* leaves contain a lectin called SteLL, which has displayed antimicrobial, immunomodulatory, antitumor, and analgesic activities. However, the effects of SteLL on the Central Nervous System (CNS) have not yet been determined. In this study, we investigated the in vivo anxiolytic effect of SteLL in mice using the open field (OF) and elevated plus maze (EPM) tests. In the OF, SteLL (1, 2, and 4 mg/kg, i.p.) did not interfere with the number of crossings but significantly reduced the number of rearings. In the EPM, SteLL 4 mg/kg and the combination SteLL (1 mg/kg) plus diazepam (1 mg/kg) significantly increased the time spent in the open arms while reducing the time spent in the closed arms. The anxiolytic effect of SteLL did not seem to be dependent on the carbohydrate-binding domain of the lectin. Nevertheless, the SteLL effect in the EPM was reversed by the pretreatment with the pharmacological antagonists of the α2-adrenoceptor, 5-HT2A/2C serotonin receptor, and the D1 dopamine receptor. Overall, our results suggest that the anxiolytic effect of SteLL is dependent on the monoaminergic signaling cascade.

## 1. Introduction

Anxiety-like disorders roughly affect more than 33% of the population, and the vast majority do not receive treatment [1]. The commonly prescribed pharmacotherapy for anxiety includes agonists of benzodiazepine receptors and inhibitors of the reuptake of serotonin, dopamine, and noradrenaline. Patients that have therapeutic opportunities have been treated with synthetic drugs that commonly elicit several side effects and lead to resistance due to their long-term use; in addition, anxiety-associated disability presents a relatively high economic cost to society. In view of this, there is an ongoing search for more sustainable, safe, and profitable drugs to treat anxiety disorders [2]. One alternative that has been gaining ground is the use of natural products.

Some natural products possess several biological and pharmacological properties that resemble the anxiolytic effect of commercialized substances. For instance, *Passiflora incarnata* Linnaeus extract, administered orally, suppressed the anxiety response in patients with spinal anesthesia, without altering the psychomotor response [3]. Additionally, a reduction in the symptoms in patients with generalized anxiety disorder was found with oral administration of the extract of *Withania somnifera* root combined with serotonin reuptake inhibitors for six weeks [4].

Lectins are proteins that have shown various biological and pharmacological potentials. These proteins present at least one carbohydrate-binding domain in their structure, which allows specific and reversible binding to carbohydrates [5,6,7]. There is not much knowledge about the effects of lectins on the Central Nervous System (CNS). Nonetheless, Russi et al. [8] showed a neuroprotective role of the lectin from *Canavalia brasiliensis* (ConBr), injected intracerebroventricularly in mice, against quinolinic acid-induced hippocampal seizures. The lectin also blocked the neurotoxicity induced by glutamate in vitro. Jacques et al. [9] confirmed the results obtained by Russi et al. [8] and demonstrated that ConBr reversed glutamate neurotoxicity via the PI3K/Akt-dependent pathway in a model of ex vivo hippocampal slices. The central administration of ConBr has also been shown to decrease the immobility time of mice in the forced swimming test, and this effect was dependent on the protein structure integrity and on the adrenergic, serotoninergic, and dopaminergic systems.

*Schinus terebinthifolia* leaf lectin (SteLL) is a glycosylated protein extracted from *S. terebinthifolia* leaf with the ability to bind chitin. The lectin has a native molecular mass of ca. 12.4 kDa [10], isoelectric point 5.7 [11], and considerable stability in terms of temperature and pH variations [11,12]. SteLL has shown biological activities such as anti-infectious [13], antimicrobial [12], and immunomodulatory [10] effects in vitro. SteLL has also demonstrated in vivo antitumor [14] and antiangiogenic [15] activities. Ramos and colleagues [16] reported an antinociceptive effect of SteLL via opioid receptors in a model of hyperalgesia in sarcoma 180-bearing mice. Recently, Marinho et al. [17] demonstrated that SteLL has peripheral and central antinociceptive action and that δ opioid receptors are involved in the antinociceptive action of SteLL against inflammatory pain. Despite this, no further data have been reported on the effect of SteLL in CNS modulation, especially concerning psychological disorders.

In this study, we evaluated the effects of the SteLL on mice models of anxiety (open field test and elevated plus maze assay) and explored whether monoaminergic signaling and the carbohydrate-binding domain of the lectin played a role in the anxiolytic efficacy of this lectin.

## 2. Results

### 2.1. SteLL Revealed an Anxiolytic Effect on the Open Field (OFT) and on the Elevated Plus Maze Test (EPM)

To evaluate the locomotive, exploratory, and anxious behavior, the animals were submitted to the OFT [18]. Figure 1A shows that the treatment of mice with SteLL (1, 2 e 4 mg/kg) did not cause significant changes in the number of crossings compared to the control. However, the treatment with the combination of SteLL (1 mg/kg) plus diazepam (0.5 mg/kg) displayed a significantly increased number of crossings when compared to the control (Figure 1A; F = 5.147, *p* = 0.0018). All groups significantly decreased the number of rearings compared to the control (Figure 1B; F = 7.698, *p* < 0.0001). Neither SteLL nor diazepam interfered with the frequency of entries (Figure 1C; F = 0.2567, *p* = 0.9330) and the time spent in the center zone of the apparatus (Figure 1D; F = 0.7328, *p* = 0.6048).

Anxiolytic-like behavior was evaluated using the EPM task [19]. Figure 2 shows the data on the evaluation of whether SteLL also promoted an anxiolytic effect in the EPM test. The treatments of mice with the highest dose of SteLL (4 mg/kg), diazepam (1 mg/kg), and SteLL (1 mg/kg) plus diazepam (0.5 mg/kg) significantly increased the time spent in the open arms compared to the control (Figure 2A; F = 10.42, *p* < 0.0001). Diazepam alone increased the number of entries into the open arms compared to the control (Figure 2B; F = 12.83, *p* < 0.0001). In addition, when analyzing the time spent by mice in the closed arm of the EPM, the data for diazepam (1 mg/kg), SteLL (4 mg/kg), and SteLL (1 mg/kg) plus diazepam (0.5 mg/kg) significantly differed from the control (Figure 2C; F = 8.109, *p* = 0.0001). Regarding the number of entries into the closed arms, a significant reduction was observed for treatments with SteLL at doses of 1 and 4 mg/kg compared to the control (Figure 2D; F = 7.491, *p* = 0.0002). Overall, the effect of SteLL was similar to the one observed in the group treated with the diazepam as positive control.

### 2.2. The Anxiolytic Effect of SteLL Did Not Depend on the Carbohydrate-Recognition Domain (CRD) of the Lectin

In another set of experiments, we evaluated whether the blockage of the CRD of SteLL with casein would revert the anxiolytic efficacy of the lectin. Nevertheless, the blockage with casein did not interfere with the time that animals spent in the open or closed arms (Figure 3A,C, respectively) nor in the number of entries into the open arms (Figure 3B) compared to the unblocked lectin, although the combined therapy (SteLL 1mg/kg and Diazepam 0.5 mg/kg) decreased the number of entries into the closed arms compared to the control (Figure 3D, F = 4.638, *p* = 0.0123). Furthermore, the blockage with casein also did not modify the effect of lectin on the crossing or rearing responses of animals in the OFT: SteLL (4 mg/kg) and diazepam (1 mg/kg) decreased the number of rearings compared to the control as much as the group administrated with SteLL 4 mg/kg blocked with casein (Figure 4B; F = 9.444, *p* = 0.0005).

### 2.3. The Anxiolytic Effect of SteLL in the EPM Is Dependent on Monoaminergic Pathways

To investigate whether the anxiolytic-like effect of SteLL in the OFT and EPM was dependent on monoaminergic pathways, the following antagonists were administered 15 min before the SteLL treatment: the nonselective antagonist of the α2-adrenoceptor (yohimbine), the 5-HT2A/2C serotonin receptor antagonist (ketanserin), or the D1 dopamine receptor antagonist (SCH 23390). Pretreatment with yohimbine, ketanserin, and SCH 23390 significantly blocked the effect of SteLL (4 mg/kg) on the time spent in the open (Figure 5A; F= 6.608 *p* < 0.0001) and closed arms (Figure 5C; F = 3.590, *p* = 0.0058) on the EPM. The number of entries into the open (Figure 5B; F = 1.089, *p* = 0.4003) or closed arms (Figure 5D; F = 1.407, *p* = 0.2383) did not significantly change among the groups. In addition, the treatment with antagonists did not affect the efficacy of SteLL in the reduction in the rearing responses on OFT (Figure 6B; F: = 7.851, *p* < 0.0001). No treatment altered the number of crossings (Figure 6A; F = 3.92, *p* > 0.05).

## 3. Discussion

In this study, the anxiolytic-like effects of lectin extracted from *Schinus terebinthifolia* leaf have been shown in the OFT and EPM tests. In addition, the data revealed that the anxiolytic effect of SteLL was dependent on the monoaminergic signaling in the EPM.

Disturbances in the monoaminergic signaling have been stated as the major force of anxiety disorders [20,21,22,23,24,25]. The use of inhibitors of monoamines reuptake has been the main treatment for anxiety, although ineffective in many patients [20,25,26]. Here, we revealed that SteLL anxiolytic-like activity in the EPM was reverted by yohimbine, ketanserin, and SCH 23390, suggesting the modulatory effects via the α2-adrenoceptor, serotonin receptor, and the D1 dopamine receptor; however, this signaling seemed not to be involved in the responses in the OFT. The EPM and the OFT are tests widely considered for measuring anxiety-like behavior in mice; nonetheless, these methodologies measure different parameters of drug-induced anxiolytic activity [27]. For instance, the EPM is based on the natural aversion of mice to open and elevated areas, whereas the OFT is based on the alterations in the exploratory behavior of animals [28,29]. In the first, an anxious behavior has its expression in the time that an animal spends in enclosed arms, while in the second the anxious behavior is mainly interpreted by the improvement in the rearing responses [30,31]. Therefore, our results suggested that the monoaminergic system involvement in SteLL efficacy was related to the aversion to light/open spaces rather than alterations in the general exploratory activity of the mice.

The interaction between the carbohydrate residues in biological membranes and lectins can modulate cell responses. For instance, Souza et al. [13] suggested that the binding of SteLL to *N*-acetylglucosamine is crucial to the lectin bacteriostatic activity. According to the literature, the interaction between carbohydrate residues and lectins can modulate cell communication and CNS neurological function [9,32,33,34]. Galectins (Gal) are endogenous lectins that belong to a family of pro- and/or anti-inflammatory endogenous β-glycan-binding proteins, and the implications of Gal-1, 3, 4, 8, and 9 in psychological and neurological diseases have been revealed [35,36,37,38]. SteLL was incubated with casein, a glycoprotein that binds its carbohydrate-binding domain, but the blockage of the CRD did not interfere with the anxiolytic efficacy of the lectin. Even though the blockage with casein reverted the number of entries into the closed arm of EPM, this isolated result does not suggest a function dependent on the lectin CRD. This is different from the result found by Marinho et al. [17], who reported that the blockage with casein inhibited the antinociceptive activity of SteLL. The results do not reject the interactions of SteLL with CNS glycans but only show that there was no need for an empty CRD for SteLL to perform the effects observed in the present work. Recently, Bezerra et al. [39] revealed that the lectin isolated from *Cratyllia mollis* seed (Cramoll 1,2,3) sustained a pesticide activity against the termite *Nasutitermes corniger* and mite *Tetranychus bastosi* independent of the carbohydrate-binding ability of the lectin.

The screening for new agents to treat anxiety disorders has been constant and yet has not been significantly successful. The screening will continue, and natural products promise to speed up the process based on their economic necessity, natural availability, and being safer alternatives. SteLL plays a flourishing role as an anxiolytic agent whose efficacy is dependent on monoaminergic signaling. Further studies must explore other signaling molecules involved in SteLL neuromodulation.

## 4. Materials and Methods

### 4.1. Animals

Male Swiss mice weighing 25–30 g, 4–5 weeks old (*n* = 72), were maintained at room temperature (22–27 °C) with free access to water and food, under a 12:12 h light–dark cycle (lights on at 7 am). All manipulations were conducted in the light phase between 9 am and 5 pm. The procedures in this study were performed according to the National Institutes of Health Guide for Care and Use of Laboratory Animals [40] and were approved by the local Ethics Committee for Animal Use (protocol no. 0010/2021) of the Federal University of Pernambuco (UFPE). All efforts were made to minimize animal suffering.

### 4.2. Purification of Lectin

Leaves of *S. terebinthifolia* were collected on the campus of UFPE in Recife, Pernambuco, according to authorization number 72024 (SISBIO) of the Chico Mendes Institute for Biodiversity Conservation (ICMBio). This research was recorded (A37C1E4) in the National System for the Management of Genetic Heritage and Associated Traditional Knowledge (SisGen). SteLL was isolated according to Gomes et al. [12]. Briefly, the flour from the dried leaves was homogenized for 16 h with 0.15 M NaCl in the proportion of 10% (w/v). After the centrifugation process (15 min, 3,500 *g*, 4 °C), the extract was loaded onto a chitin (Sigma-Aldrich, St. Louis, MO, USA) column equilibrated with 0.15 M NaCl. The adsorbed proteins (SteLL) were eluted with 1.0 M acetic acid and dialyzed against distilled water for 6 h. The protein concentration was determined according to Lowry et al. [41] using the bovine serum albumin standard curve (31.25–500 µg/mL). The carbohydrate-binding ability of the SteLL samples was determined through a hemagglutinating activity assay as described by Procópio et al. [42].

### 4.3. Drugs and Treatment

To evaluate the anxiolytic efficacy of SteLL, mice received intraperitoneally the lectin at 1, 2, or 4 mg/kg (in phosphate-buffered saline, PBS; *n* = 6 per group) or only the vehicle PBS (control; *n* = 6) 30 min before the open field (OF) and elevated plus maze (EPM) tests. Lectin doses were defined based on Ramos et al. [16], who found antinociceptive action of SteLL at 1 and 2 mg/kg. Diazepam (1 mg/kg i.p.) was used as a positive control (*n* = 6).

To determine the monoaminergic signaling involvement in SteLL anxiolytic activity, the nonselective antagonist of the α2-adrenoceptor (yohimbine; 1 mg/kg), the 5-HT2A/2C serotonin receptor antagonist (ketanserin; 5 mg/kg), or the D1 dopamine receptor antagonist (SCH 23390; 0.05 mg/kg) were administrated i.p. 15 min before SteLL (4 mg/kg) administration. Six animals were assigned to each group. The choice and the doses of the antagonists were based on Araujo et al. [43], and the drugs were also dissolved in PBS. All drugs were obtained from Sigma-Aldrich and were prepared fresh on the day of testing. After administration, the animals were evaluated in the EPM and OF tests.

In addition, a combination of SteLL at 1 mg/kg and a subeffective dose of diazepam (0.5 mg/kg) was administered 15 min before EPM and OF tests to assess the efficacy of combined therapy (*n* = 6).

### 4.4. The Effect of the Carbohydrate-Recognizing Domain (CRD) on the Anxiolytic Effect of the Lectin

To determine whether the effect of the lectin was dependent on the CRD of the lectin, the CRD was blocked by dissolving the lectin (4 mg/kg) in a PBS buffer containing 0.5 mg/kg of casein (a glycoprotein that inhibits SteLL hemagglutinating activity), and the solution (SteLL + casein) was kept at 37 °C for 30 min before being administered to the animals (n = 6). The control group received only casein (*n* = 6). The OFT and EPM were then performed 30 min after treatments.

### 4.5. Open Field (OF) Test

The OF apparatus consisted of a wooden box measuring 40 × 60 × 50 cm [18]. The floor of the arena was divided into twelve equal squares. The number of squares crossed with all paws (crossing) and the number of times the animal was supported only on its hind legs (rearing) were counted in a 5-min session [44]. Each animal was placed individually and carefully on the apparatus. The apparatus was sanitized between each animal so that there was no interference of smells. The anxiolytic activity was recognized through the alteration in the rearing responses of mice and the alterations in the number of crossings [45].

### 4.6. Elevated Plus Maze (EPM) Test

Anxiolytic-like behavior was evaluated using the EPM task as previously described [19]. The EPM consisted of two opposite open arms (50 × 10 cm), crossed with two closed walls of the same dimensions, with 40 cm high walls. The arms were connected with a central square of 10 × 10 cm and the entire maze was placed 50 cm above the ground. The animals were placed on the central platform of the maze in front of an open arm. The animal had 5 min to explore the apparatus, and the time spent and the number of entries into open and closed arms were recorded [46]. Each animal was placed individually and carefully in the center, facing the open arm. The apparatus was thoroughly cleaned with 30% ethanol between each session. An increase in the time spent in the closed arms by mice was interpreted as anxiolytic behavior.

### 4.7. Statistical Analysis

Comparisons between the experimental and control groups were performed by one-way analysis of variance (ANOVA) followed by the Bonferroni test. A value of *p* < 0.05 was considered significant. All statistical procedures were carried out using PRISMA Statistic software version 8.0.

## 5. Conclusions

Overall, our results indicate that SteLL plays a flourishing role as an anxiolytic agent whose efficacy is dependent on monoaminergic signaling but not on the lectin CRD. Further studies must explore other signaling molecules involved in SteLL neuromodulation as well as the effects of this lectin on the psychological disorders.

## Figures and Tables

**Figure 1 pharmaceuticals-15-01364-f001:**
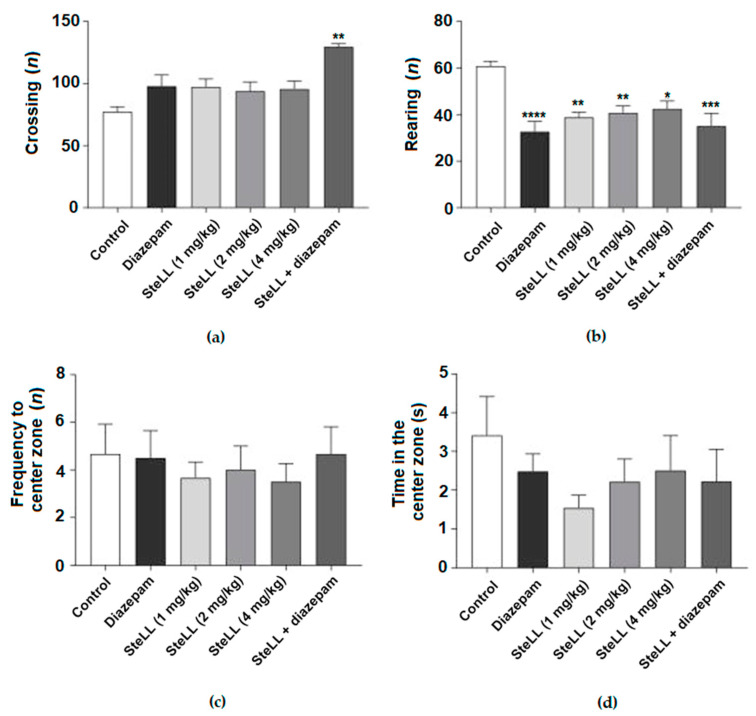
Effects of SteLL (1, 2, and 4 mg/kg) i.p. treatment in the open field test. Diazepam (1 mg/kg) was used as a positive control. Diazepam (0.5 mg/kg) plus SteLL (1 mg/kg) was used to evaluate the synergic effect of both compounds combined. (**a**) Total number of crossings. (**b**) Total number of rearings. (**c**) Frequency of the mice entering the center zone. (**d**) The overall duration of time mice spent in the center of the apparatus. Significant differences compared with the control group: (*) *p* < 0.05; (**) *p* < 0.001; (***) *p* < 0.0001; (****) *p* < 0.00001.

**Figure 2 pharmaceuticals-15-01364-f002:**
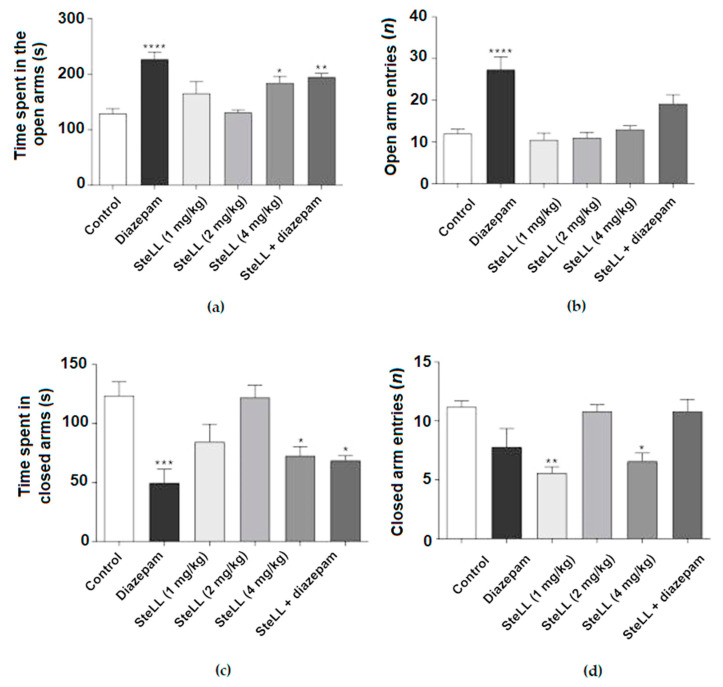
Effects of SteLL (1, 2, and 4 mg/kg) i.p. treatment on the elevated plus maze test. Diazepam (1 mg/kg) was used as a positive control. Diazepam (0.5 mg/kg) plus SteLL (1 mg/kg) was used to evaluate the synergic effect of both compounds combined. (**a**) Total time that mice spent in open arms. (**b**) The number of entries into the open arms. (**c**) Total time spent in closed arms. (**d**) The number of entries into the closed arms. Significant differences compared with the control group: (*) *p* < 0.05; (**) *p* < 0.001; (***) *p* < 0.0001; (****) *p* < 0.00001.

**Figure 3 pharmaceuticals-15-01364-f003:**
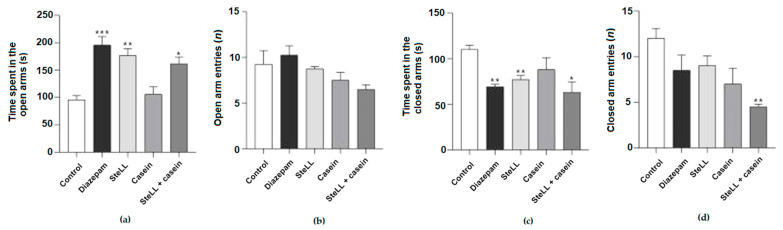
The incubation of the lectin with casein did not revert the anxiolytic effect of SteLL (4 mg/kg) in the elevated plus maze test. (**a**) Total time that mice spent in open arms. (**b**) The number of entries into the open arms. (**c**) Total time spent in closed arms. (**d**) The number of entries into the closed arms. (*) *p* < 0.05; (**) *p* < 0.001; (***) *p* < 0.0001 compared to control.

**Figure 4 pharmaceuticals-15-01364-f004:**
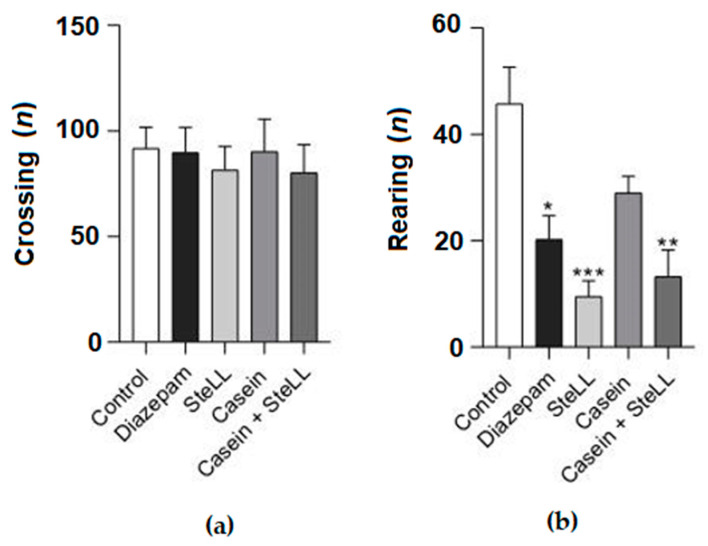
The incubation of the lectin with casein did not revert the anxiolytic effect of SteLL (4 mg/kg) in the open field test. (**a**) Total number of crossings. (**b**) Total number of rearings. (*) *p* < 0.05; (**) *p* < 0.001; (***) *p* < 0.0001 compared to control.

**Figure 5 pharmaceuticals-15-01364-f005:**
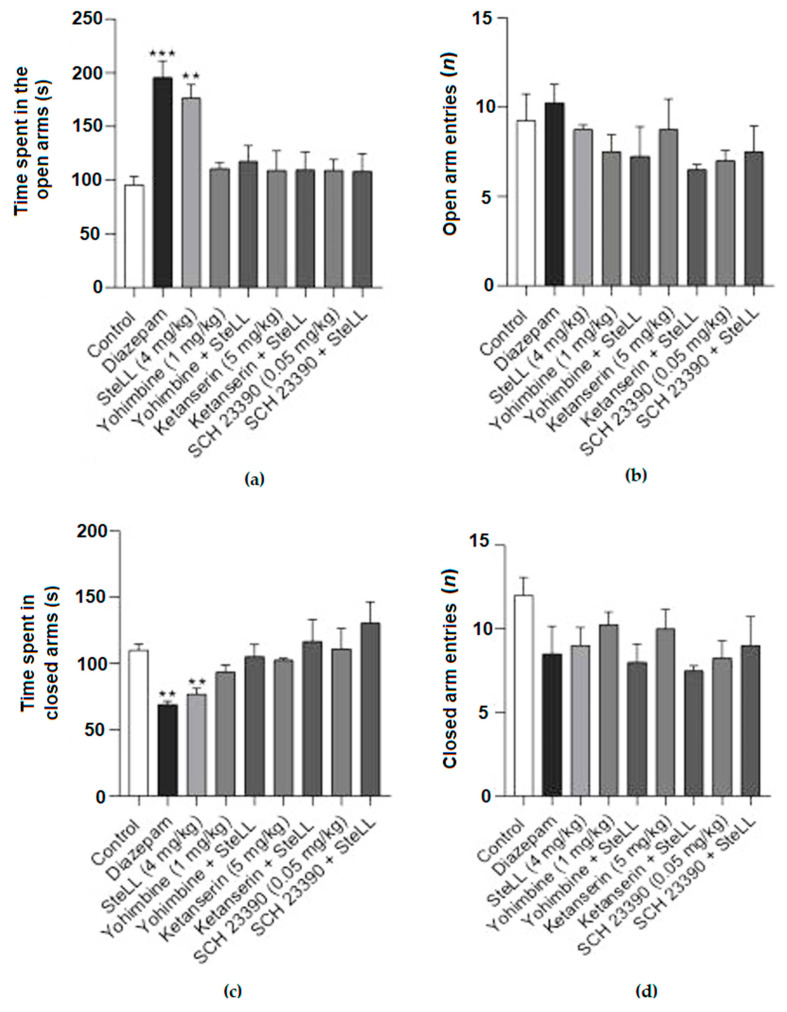
The anxiolytic-like effect of SteLL in the EPM was dependent on monoaminergic signaling. The pretreatment with the nonselective antagonist of the α2-adrenoceptor yohimbine 1 mg/kg, the 5-HT2A/2C serotonin receptor antagonist ketanserin 5 mg/kg, and the D1 dopamine receptor antagonist SCH 23390 0.05 mg/kg inhibited the effect of SteLL (4 mg/kg) on the time spent in the open (**a**) and closed (**c**) arms. No changes were observed in the number of entries in open (**b**) and closed (**d**) arms. The antagonists were administrated 15 min before SteLL administration. The experiments were conducted 15 min after SteLL administration. (**) *p* < 0.001; (***) *p* < 0.0001 compared to control.

**Figure 6 pharmaceuticals-15-01364-f006:**
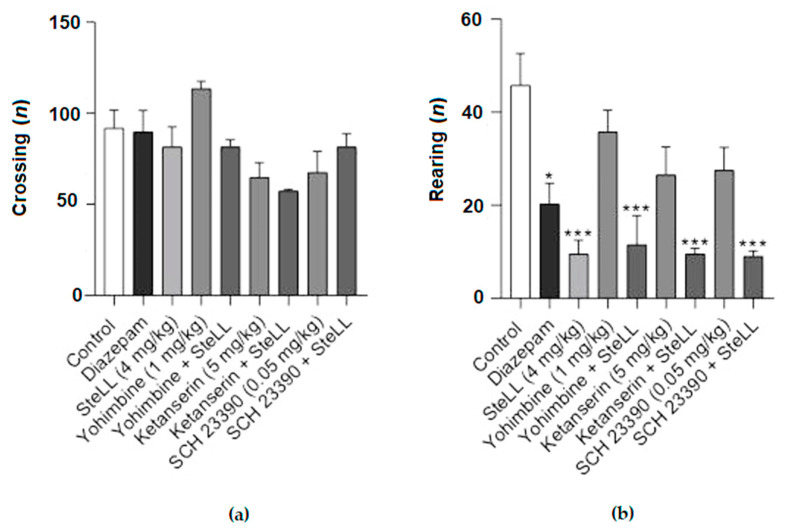
The blockage of the monoaminergic signaling did not alter the rearing responses of SteLL administration (4 mg/kg) on the OFT. The nonselective antagonist of the α2-adrenoceptor yohimbine 1 mg/kg, the 5-HT2A/2C serotonin receptor antagonist ketanserin 5 mg/kg, and the D1 dopamine receptor antagonist SCH 23390 0.05 mg/kg were evaluated. (**a**) There were no significant changes in the numbers of crossing among the groups. (**b**) SteLL 4 mg/kg decreased the number of rearings compared to the control even in animals pretreated with the antagonists. (*) *p* < 0.05; (**) *p* < 0.001; (***) *p* < 0.0001 compared to control.

## Data Availability

Data is contained within the article.

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
