# Peer review of "The Anxiolytic Activity of Schinus terebinthifolia Leaf Lectin (SteLL) Is Dependent on Monoaminergic Signaling although Independent of the Carbohydrate-Binding Domain of the Lectin"

_pharmaceuticals, 2022, doi:10.3390/ph15111364_

Round 1
Reviewer 1 Report
The authors identified anxiolytic activity of Schinus terebinthifolia leaf lectin (SteLL) in mice models. The SteLL played a healthy role as an anxiolytic agent whose efficacy is dependent on monoaminergic signaling. The article needs the following changes:
Keywords: please rearrange in alphabetical order.
The conclusion part needs to be moved immediately after the discussion part.
The paper’s content is interesting, but the present form of the article needs some revisions.
Line 30-31: need to be rewritten.
References: Please add in the methods section; also in lines 169, and 226.
Revise the statements in introduction and discussion part. Overall several grammatical and incomplete sentences have been found. Please revise/rewrite.
Author Response
- Keywords were rearranged in alphabetical order.
-
Conclusion was placed immediately after the discussion part.
-
Line 30-31: the text was corrected.
-
Reference for line 169 is the same cited in line 171. The text was adjusted.
- Reference for line 226 was included (line 232 in revised version).
-
The text was revised.
Reviewer 2 Report
Manuscript is interesting. Authors aimed to assess the anxiolytic activity of lectin isolated from Schinus terebinthi leaf. The study is within important trend to find new drug of plant origin. Introduction gives relevant background and highlights the significance of the study. The results are clearly presented and methodology is correct. I have only some suggestions before acceptance for publication.
1) Line 45: incorrect format for reference (Fuladi et al., 2021).
2) Figures should be moved near the place where they are cited in the text. For example, Fig.2 is described in 2.1 section and it is placed in 2.2 (the same comment for Fig. 5 and 6)
3) 4.1. section: lack of number of animal used in the study
4) 4.3. section: lack of number objects in particular group. Scheme of experimental design could be added.
Author Response
- Fuladi et al. (2021) was corrected and added to the reference list.
- Figures were put as near as possible of the text, but according to the template, we need to avoid blank spaces in the pages. In addition, we are confident that the editorial team will modify the position and size of the figures if they deem it necessary.
- Number of animals were included
- Number objects in each group were detailed.
Reviewer 3 Report
The manuscript presents the in vivo evaluation of the anxiolytic-like activity of lectin from Schinus terebinthifolia leaves (SteLL). For this purpose, the authors used the open field (OF) and elevated plus maze (EPM) tests. Results showed that in the OF, SteLL (1, 2, 18, and 4 mg/kg, i.p.) did not interfere with the number of crossings but significantly reduced the number of rearing. But in the EPM, SteLL 4 mg/kg and the combination SteLL (1 mg/kg) plus diazepam (1 20 mg/kg) significantly increased the time spent in the open arms while reducing the time spent in the 21 closed arms. The anxiolytic effect of SteLL did not seem to depend on the lectin's carbohydrate-binding domain. The pre-treatment reversed the SteLL effect in the EPM with a wide panel of pharmacological antagonists. The authors concluded that the anxiolytic effect of SteLL is dependent on the monoaminergic signaling cascade. Generally, the manuscript is presented well-structured, and the experiments are designed appropriately to test the hypothesis.
However, the Reviewer found some concerns regarding the doses of SteLL used for experiments. It must be specified and precise in the manuscript how the authors have decided to select presented in the manuscript the doses of SteLL. Moreover, how the authors chose the receptor antagonists? Why yohimbine and SCH23390 were used? It must be specified and precise in the manuscript.
Minor: lack of Fudali et al. in the reference list, sarcoma 180 – better to change to sarcoma 180-bearing mice
Author Response
- The explanation for the doses of SteLL used were already provided in the original version (lines 248-249). However, we expanded the information in the revised version.
- Yohimbine is a well-known alpha-2 adrenoreceptor antagonist used in several assays to study anxiety. SCH 23390 has been a major tool for studies on D1 receptor involvement in neurological function and dysfunction. The text was altered to: "The choice and the doses of the antagonists were based on Araujo et al. [43]".
- Fuladi et al. (2021) was adjusted and included in the list of references.
- The text was modified to: "antinociceptive effect of SteLL via opioid receptors in a model of hyperalgesia in sarcoma 180-bearing mice"